# HOPE-SIM, a cryo-structured illumination fluorescence microscopy system for accurately targeted cryo-electron tomography

Shuoguo Li [1,2,3], Xing Jia[1], Tongxin Niu[1], Xiaoyun Zhang[1], Chen Qi[1], Wei Xu[1], Hongyu Deng [3,4,5], Fei Sun [1,2,3,5 ✉] & Gang Ji [1,3 ✉]

Cryo-focused ion beam (cryo-FIB) milling technology has been developed for the fabrication of cryo-lamella of frozen native specimens for study by in situ cryo-electron tomography (cryo-ET). However, the precision of the target of interest is still one of the major bottlenecks limiting application. Here, we have developed a cryo-correlative light and electron microscopy (cryo-CLEM) system named HOPE-SIM by incorporating a 3D structured illumination fluorescence microscopy (SIM) system and an upgraded high-vacuum stage to achieve efficiently targeted cryo-FIB. With the 3D super resolution of cryo-SIM as well as our cryo-CLEM software, 3D-View, the correlation precision of targeting region of interest can reach to 110 nm enough for the subsequent cryo-lamella fabrication. We have successfully utilized the HOPE-SIM system to prepare cryo-lamellae targeting mitochondria, centrosomes of HeLa cells and herpesvirus assembly compartment of infected BHK-21 cells, which suggests the high potency of the HOPE-SIM system for future in situ cryo-ET workflows.

[1] Center for Biological Imaging, Core Facilities for Protein Science, Institute of Biophysics, Chinese Academy of Sciences, 100101 Beijing, China. [2] National Key Laboratory of Biomacromolecules, Institute of Biophysics, Chinese Academy of Sciences, 100101 Beijing, China. [3] University of Chinese Academy of Sciences, 100049 Beijing, China. [4] CAS Key Laboratory of Infection and Immunity, Institute of Biophysics, Chinese Academy of Sciences, 100101 Beijing, China. [5] CAS Center for Excellence in Biomacromolecules, Institute of Biophysics, Chinese Academy of Sciences, 100101 Beijing, China. ✉email: feisun@ibp.ac.cn; jigang@ibp.ac.cn

Revealing three-dimensional (3D) cellular ultrastructures is an important step in understanding the life. In recent years, cryo-electron microscopy (cryo-EM) has become increasingly improved and is becoming one of the major biophysical techniques used to study high-resolution 3D structures of macromolecular complexes[1]. In addition, cryo-electron tomography (cryo-ET) has emerged as another powerful tool to visualize the macromolecular organization of unperturbed cellular landscapes with the potential to attain near-atomic resolution[2–4]. To visualize the subcellular ultrastructure in vitrified cells by cryo-ET, focused ion beam milling under cryogenic conditions (cryo-FIB) has been used to prepare thin cryo-lamellae from vitrified cells, enabling many exciting biological observations inside cells. The three-dimensional organization of cellular organelles, such as the cytoskeleton[5–9], endoplasmic reticulum (ER)[10], and 26S proteasome[11] have been successfully analyzed by cryo-ET.

There are several limitations that preclude the wider application of cryo-ET, including difficulties in locating and identifying features of interest[12]. Cryo-correlative light and electron microscopy (cryo-CLEM)[13–18] has been proven to be an effective approach to overcoming this problem by utilizing fluorescent labeling to navigate toward the target for the subsequent cryo-FIB milling of vitrified cells[14–17] or cryo-ET imaging of frozen-hydrated sections[12,18]. Considering that the normal thickness of cells is far beyond the mean free path of 300 keV electrons, the fabrication of vitrified cells to a thickness of ~200 nm is necessary before cryo-ET data collection. Therefore, the sequential experimental procedure from vitrification, cryo-fluorescence microscopy (cryo-FM), cryo-FIB, and cryo-ET has become a routine workflow for many site-specific in situ structural studies[12,19–23]. This workflow normally requires a stand-alone fluorescence microscope with a cryo-stage for cryo-fluorescence imaging[15–17,24,25], followed by cryo-scanning electron microscopy (cryo-SEM). A correlation alignment between the cryo-FM and cryo-SEM images is generated and used to guide cryo-FIB milling[12,18,21,26]. In addition, specific fiducial markers imaged by both cryo-FM and cryo-FIB, such as fluorescent beads, are required for correlation alignment using specific 3D correlative software[21,27]. There have been many hardware implementations reported to realize this workflow by developing cryogenic wide-field fluorescence microscopy[13,21,22] or high-resolution cryo-Airyscan confocal microscopy (CACM)[12].

The correlation accuracy and success rate between cryo-FM and cryo-FIB is limited by the resolution of cryo-FM and further attenuated by the factors of specimen deformation, devitrification, and ice contamination[7,21]. We previously developed a unique high-vacuum optical platform for cryo-CLEM (HOPE)[25], which utilized an integrative cryo-holder and a specific vacuum chamber to minimize ice contamination and reduce the risk of specimen deformation and devitrification during cryo-FM imaging and specimen transfer. However, the use of a wide-field fluorescence microscope with a low numeric aperture (NA) objective lens and without the information of the z position of fluorescent targets in our HOPE system has limited the correlation accuracy above one micron, which needs to be improved to increase the success rate of site-specific cryo-FIB milling.

Structured illumination microscopy (SIM) introduces patterned illumination light on the imaging target, resulting in low-frequency interference fringes that carry structural information beyond the diffraction limit of the system, which doubles the resolution of conventional wide-field fluorescence microscopy and enables the visualization of detailed molecular processes in the whole cell[5,28,29]. Compared to other superresolution FM techniques, such as photoactivated localization microscopy (PALM)[30], stochastic optical reconstruction microscopy (STORM)[31], stimulated emission depletion (STED) nanoscopy[32] and 4Pi microscopy[33], SIM uses relatively low illumination power (10–100 mW cm$^{-2}$) and requires no special fluorophores or objective lenses to double the diffraction-limited resolution in three dimensions[28,34–38]. The low illumination power minimizes the heating effect on the specimen, which is important for avoiding the risk of devitrification[39,40] for frozen-hydrated specimens. Therefore, SIM has been recently employed within correlative microscopy with soft X-ray microscopy to visualize cells in vitreous ice[35].

In this work, starting from our previously developed HOPE system[25], we developed a cryo-correlative light and electron microscopy (cryo-CLEM) system with the name HOPE-SIM to achieve efficiently targeted cryo-FIB. We upgraded the wide-field fluorescence microscope to a 3D-SIM system to increase the FM imaging resolution in the x and y dimensions and obtain additional information in the z dimension, which greatly improves the correlation accuracy to guide site-specific cryo-FIB fabrication. We also upgraded the high-vacuum system to improve the vacuum of the chamber and further reduce the rate of ice growth, which allows a longer time of cryo-FM imaging before the specimen undergoes frosting. We also developed a specific 3D correlative software, 3D-View, to perform the fiducial marker-based correlation between cryo-SIM and cryo-FIB images, which is used to navigate cryo-FIB milling accurately.

We measured the precision of targeting region of interest, which can reach to 110 nm. And we successfully applied HOPE-SIM-based cryo-CLEM workflow to target mitochondria, herpesvirus assembly compartment of infected BHK-21 cells, and centrosomes of HeLa cells.

## Results

**Design of the HOPE-SIM system.** Our HOPE-SIM system (Figs. 1 and 2a, Supplementary Fig. 1, and Supplementary Movie 1) was upgraded from our previously developed HOPE system[25]. First, a larger high-vacuum chamber was designed and mounted onto a commercial epi-fluorescence microscope (e.g., model IX73, Olympus Corporation, Japan) with a high numerical aperture (NA) long working distance dry objective lens (e.g., Nikon CFI TU Plan Apo EPI ×100, NA/WD: 0.9/2 mm) mounted in the chamber. Placing the objective lens inside the vacuum chamber allows for the utilization of a high NA lens to increase the FM resolution. Like our previous design, a vacuum transfer system with a pre-pumping device is connected to the vacuum chamber via a bellow tube and a manual gate valve, which allows a commercial cryo-holder to be transferred into the vacuum chamber and the cryo-specimen to be placed above the objective lens in the light path. The movement of the cryo-holder is driven by a three-axis electric motor stage that is fixed to the vacuum chamber. Different from our HOPE system, an inserted anti-contamination system (ACS) was designed, fixing a liquid nitrogen dewar inside the vacuum chamber to cool a cryo-box (Fig. 2a and Supplementary Fig. 1b). This cryo-box with a working temperature lower than −175 °C is used to cage the tip of the cryo-holder (Supplementary Fig. 1c–e) and protect the cryo-specimen from ice deposition/contamination. In addition, there are two ports designed on the left of the vacuum chamber, with one port connected to a turbo pump system (TPS) and another connected to a vacuum gauge (Supplementary Fig. 1a, b). A high-power TPS is used to achieve an improved vacuum better than $5 \times 10^{-5}$ Pa under cryogenic conditions. We designed a touch screen device to monitor the vacuum levels of both the chamber and the gate valve, the pumping status, and the temperature of the cryo-box (Supplementary Fig. 1a). To monitor the system around the objective lens inside the chamber, we designed an observation glass window in front of the chamber (Supplementary Fig. 1a, b, e). To allow the light to pass through the vacuum chamber, there are two glass optical windows designed on the top and bottom of the objective lens (Supplementary Fig. 1f, g).

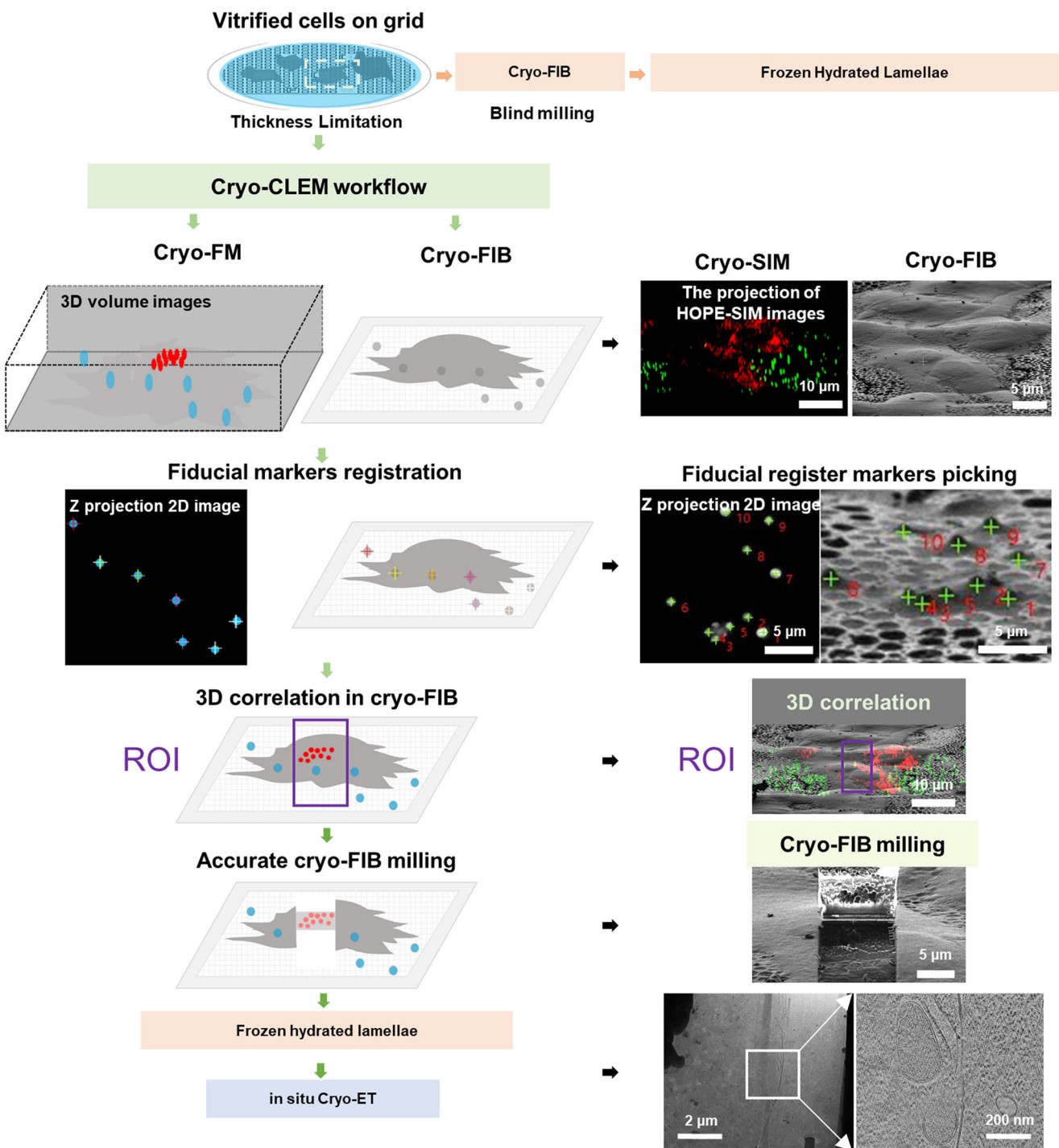

**Fig. 1 Workflow of cryo-CLEM with one example using cryo-SIM.** Vitrified cells on EM grids are too thick to be imaged directly by cryo-EM. Blind milling by cryo-FIB lacks the precision to target the region of interest (ROI), which needs to be solved by cryo-CLEM. The vitrified cells were first imaged by cryo-FM and screened with good fluorescent signals. The fluorescent images of both microspheres (blue) and ROIs were captured. The fluorescent images are then correlated with cryo-FIB images by aligning the positions of microspheres (Fiducial marker registration). The precise correlation is then used to guide accurate cryo-FIB milling to prepare target cryo-lamellae of the ROI for the subsequent in situ cryo-ET study. An example using cryo-SIM of the HOPE-SIM system to demonstrate the whole workflow of cryo-CLEM is shown in the right panels. HeLa cells stained with MitoTracker Red CMXRos (red channel) were used for this demo. The fluorescent microspheres were imaged in the green channel and correlated with the cryo-FIB image, where the microspheres are marked with crosses. Accurate cryo-FIB milling was performed with the guidance of 3D correlation, and the subsequent cryo-ET reconstruction showed the location of the target mitochondria in situ.

To avoid touching the grid directly in the subsequent cryo-FIB and cryo-ET, we selected FEI AutoGrid for cryo-specimen transfer during the whole cryo-CLEM workflow. A multi-specimen single tilt cryo-holder (Model 910.6, Gatan, USA) was used to transfer the

cryo-specimen into the vacuum chamber and maintain the cryogenic condition during cryo-FM imaging. To fit AutoGrid with this commercial cryo-holder, we used our previously reported holder tip[41] (Supplementary Fig. 1c, d) to hold AutoGrid

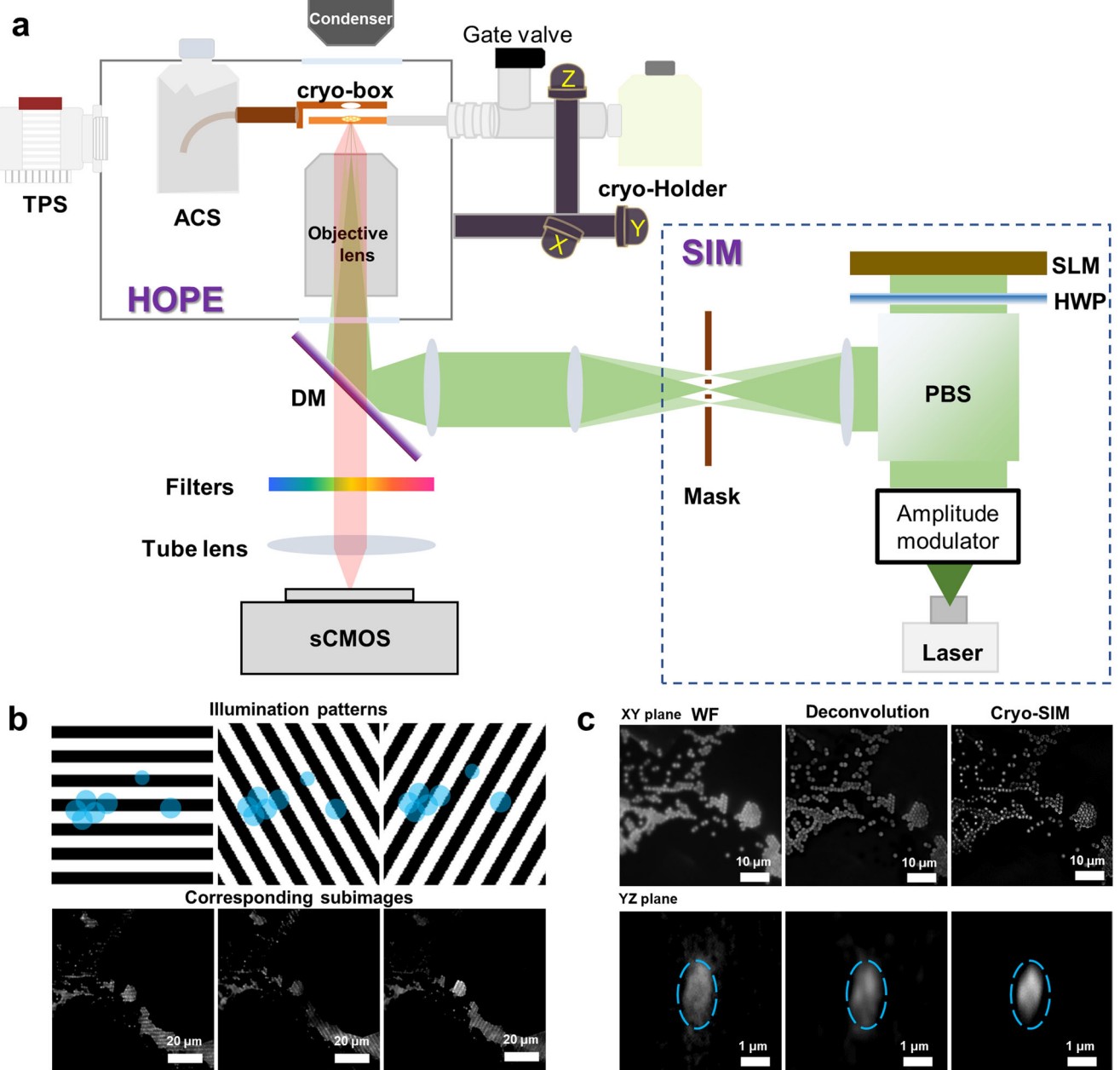

**Fig. 2 Design and principle of the HOPE-SIM system. a** Schematic overview of the HOPE-SIM system design in its operational mode. Each part of the system is labeled and described in the text. A structured illumination beam is induced into the microscope via the backlight inlet. TPS turbo pump system, ACS anti-contamination system, DM dichroic mirror, SIM structured illumination microscopy, SLM spatial light modulator, HWP half-wave plate, PBS polarizing beam splitter. **b** Diagrammatic (top) and real (bottom) corresponding subimages of the cryo-SIM illumination patterns (black and white) on the fluorescence microspheres (blue). **c** Comparisons of fluorescent images of 1-μm microspheres among different imaging modes: wide-field (WF), deconvolution and cryo-SIM. It is clear that cryo-SIM yields the image with the best resolution in both the *XY* and *YZ* planes.

mounted with our D-shaped finder grids[42] (Supplementary Fig. 1c). This special holder tip is important in protecting cryo-specimens from damage and providing enough mechanical stability for subsequent cryo-FM imaging.

In addition to upgrading the vacuum system and mechanics, we also upgraded the wide-field fluorescence microscope of our HOPE system to SIM (Fig. 2 and Supplementary Fig. 2), which utilizes a laser source with three channels (561 nm/488 nm/ 405 nm) and a spatial light modulator (SLM, QXGA-3DM, Forth Dimension Displays) to generate a structured illumination pattern. The resulting fluorescence signals are collected by the objective and recorded on a high-sensitivity sCMOS camera

(Prime 95B, Photometrics). 3D-SIM raw data can be obtained to reconstruct a stack of fluorescence image series that will be used for the subsequent correlation and cryo-FIB navigation.

To control the whole system (laser, SLM, stage, and camera) of HOPE-SIM and be compatible with the subsequent correlation, we designed a LabVIEW (National Instrument, USA)-based software, HOPE-SIM View (Supplementary Fig. 3). HOPE-SIM View can be used to set up cryo-FM imaging parameters (e.g., the exposure time and focusing zone), map the area of the whole grid, register the list of targets of interest (Supplementary Fig. 3a), set exposure time (Supplementary Fig. 3b), change the laser channel and power intensity (Supplementary Fig. 3c), control the stage

movement (Supplementary Fig. 3d), and collect multichannel 3D-SIM raw data.

**HOPE-SIM-based cryo-CLEM workflow.** Using our HOPE-SIM system, we can incorporate cryo-SIM imaging into the current standard protocol of cryo-CLEM (Fig. 1). In detail, the D-shaped finder grid is used to grow cells to a suitable density, and the cells are then either transfected to express target florescent tagged molecules or stained with fluorescent dyes. Before plunge freezing, a proper amount of diluted fluorescent microspheres with a recommended size of 1 micron in diameter are added into the culture on the grid. The vitrified grid is then assembled into AutoGrid and mounted into our specially designed tip that is fitted into the cryo-multi-holder for the subsequent cryo-SIM imaging. Notably, the side of the grid with the growing cells should face to the objective lens.

Before cryo-SIM imaging, the vacuum chamber of the HOPE-SIM system must be pumped to a high vacuum of $\sim 5 \times 10^{-5}$ Pa, and the ACS needs to be cooled to less than $-170\,°C$ by liquid nitrogen, which is important to reduce the ice growth rate during cryo-SIM imaging (Supplementary Fig. 4 and Supplementary Data 1 and 8). The cryo-multi-holder is then inserted into the HOPE-SIM system by the vacuum transfer device. After $\sim 5$ min of waiting time to recover the vacuum of the chamber and another $\sim 5$ min for the temperature of the holder tip to stabilize, the shutter of the cryo-multi-holder is open and ready for sample screening and cryo-SIM imaging. The frozen grid is first screened in transmission wide-field imaging mode, and the concentration of cells, positions of cells in the square, integrity of the supporting film, ice contamination and thickness of the specimen are checked. The regions with cells showing a good fluorescent signal at the right position in the square, an unbroken supporting film, an appropriate ice thickness, and well-dispersed fluorescent beads are selected and registered as ROIs (regions of interest) in HOPE-SIM View for the subsequent 3D cryo-SIM data collection. The field of vision of HOPE-SIM system, about 140 microns, exactly covers a whole square of a 200-mesh grid. And the multichannel 3D imaging is allowed to capture both fiducial markers and target fluorescent molecules.

After cryo-SIM imaging, the frozen grid assembled with AutoGrid is transferred into a dual beam cryo-SEM (e.g., FEI Helios NanoLab 600i) via our previously developed protocol and cryo-shutter[42]. We developed LabVIEW (National Instrument, USA)-based software, 3D-View (Supplementary Fig. 5), for image correlation between different microscopes (Supplementary Movie 2). First, the whole grid is screened by cryo-FIB imaging in low magnification, and the positions of the ROIs can be found via the index of the Finder. The ROIs with cells having good shape and suitable density are selected for subsequent cryo-FIB imaging with a proper magnification of $\times 2500$ (Supplementary Fig. 5a, b). Second, the positions of fluorescent beads from both the cryo-SIM and corresponding cryo-FIB images are selected for precise correlation using 3D-View (Supplementary Fig. 5b) (the detailed algorithm is described in Methods). After image correlation, the optimized Euler angle and minimized standard deviation (SD) can be obtained and displayed (Supplementary Fig. 5c–e). Finally, the cryo-SIM image can be merged with the cryo-FIB image (Supplementary Fig. 5f) to highlight the target for the subsequent cryo-FIB fabrication.

Using AutoGrid, the prepared cryo-lamella is then ready for the cryo-ET experiment using a 300 kV cryo-electron microscope (e.g., FEI Titan Krios). The cryo-SIM image is also useful to help locate the ROI for cryo-ET data collection.

**Precision of targeted cryo-FIB.** Using the protocol of the open-source software SIM-Check[43], we measured the resolution

and checked the stability/quality of our cryo-SIM system (Supplementary Fig. 2d–i). As a result, an $\sim 200$ nm lateral resolution and an $\sim 500$ nm $z$-direction resolution was achieved (Supplementary Fig. 2e, f), which are better than the wide-field image as well as the deconvolution image (Fig. 2c). The enhanced resolution and signal-to-noise ratio improved the accuracy of the determined $z$-coordinates of the fluorescent microspheres, which is important for the subsequent correlation accuracy between the 3D cryo-SIM and 2D cryo-FIB images.

We then measured the targeting precision of cryo-FIB guided by the 3D correlation information in 3D-View. In our first measurement (Fig. 3a–e), we spread two differently sized and colored fluorescent microspheres (Green/0.5 μm, 505/515 nm, F8813, Thermo Fisher Scientific Corp., OR., red/0.2 μm, 580/605 nm, F8810, Thermo Fisher Scientific Corp., OR.) onto a GraFuture RGO TEM Grid (GraFuture™-RGO001A, Shuimu BioSciences Ltd., CN) and left it dry. This grid was used to measure the correlation precision between cryo-SIM and cryo-FIB images. We recorded both green and red channels of cryo-SIM images (Fig. 3a), selected the channel of green fluorescent microspheres to correlate the 3D cryo-SIM and 2D cryo-FIB images (Fig. 3b–d), and then calculated the standard deviation of the red fluorescent microspheres (see Eq. (8) in "Methods"), resulting in a correlation precision of 110 nm (Fig. 3e and Supplementary Data 2).

In our second measurement (Fig. 3f–l), to mimic the real experimental procedure of target cryo-FIB milling, we vitrified the mixed solution of 1 μm blue microspheres (430/465 nm, F13080, Thermo Fisher Scientific Corp., OR) and 5 μm polymethyl methacrylate beads (PMMA5, Kai New Materials Ltd., CN) directly on the EM grid. With the shielding of big polymethyl methacrylate beads, many blue microspheres were embedded in the vitrified ice, mimicking the fluorescent target embedded in the vitrified cell. We then selected five embedded microspheres (B1-5 in Fig. 3f) to test the precision of the target cryo-FIB milling. First, we utilized the un-embedded blue microspheres that were discernible in cryo-FIB image to correlate the 3D cryo-SIM and 2D cryo-FIB images (Fig. 3g, h). The correlation image (Fig. 3h) showed that the selected microspheres were unrecognizable and exactly close to the big polymethyl methacrylate beads according to the correlated fluorescent signals. Then cryo-FIB milling was performed at these target position, respectively (Fig. 3i, j). The resulted cryo-lamella with a thickness of $\sim 200$ nm was then imaged by SEM at a high magnification (Fig. 3k). The diameters of milled target microspheres were measured (Fig. 3l and Supplementary Fig. 6), resulting an averaged diameter measurement of $908 \pm 106$ nm (Supplementary Data 3). Compared to the statistical measurement of diameters of the blue microspheres in cryogenic condition (Supplementary Fig. 7 and Supplementary Data 4), which showed the averaged diameter of the blue microspheres was $997 \pm 32$ nm, the precision of target cryo-FIB milling in our workflow is at least smaller than 100 nm.

Notably, we found that the distribution of the fluorescent microspheres in the $y$ direction (perpendicular to the FIB scanning direction) and the precision of determination of the $z$-coordinates of the microspheres are key factors affecting the accuracy of the correlation (see also Supplementary Note 1).

**Application of in situ ultrastructural study.** We applied our HOPE-SIM-based cryo-CLEM workflow to study real biological systems. The first test was to target the mitochondria of HeLa cells that were stained with MitoTracker Red CMXRos (Invitrogen M7512, Thermo Fisher Scientific Corp., OR.). With the improved resolution of cryo-SIM, the mitochondria can be well resolved and separated in the fluorescent image, and the

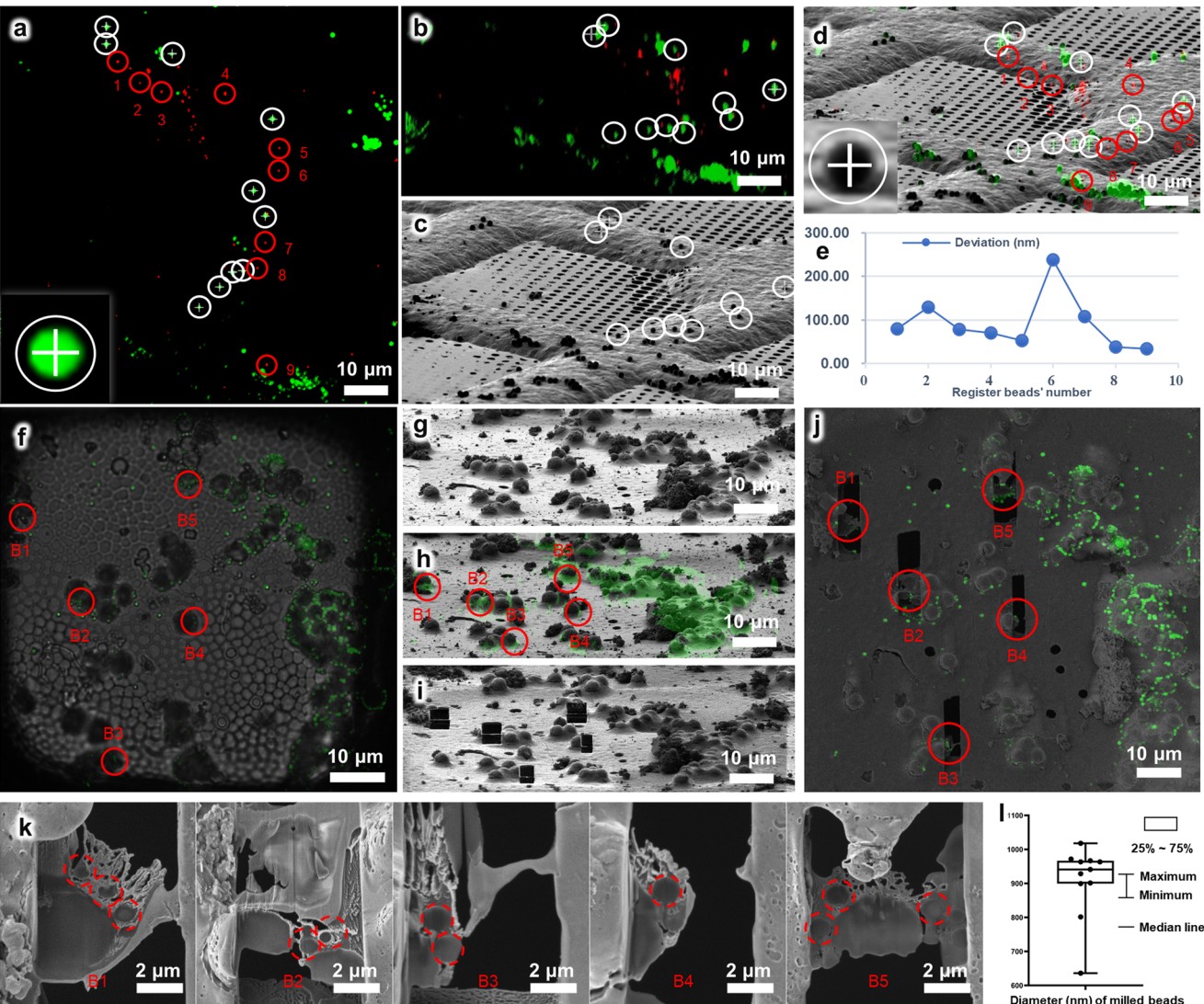

**Fig. 3 Correlation accuracy of the HOPE-SIM system. a** HOPE-SIM images of dried 0.5 μm green and 0.2 μm red fluorescent microspheres spread on a GraFuture RGO grid. The green microspheres tagged by white circles and crosses are used as the fiducial register markers. The red microspheres labeled from 1 to 9 are used to measure the correlation accuracy with the FIB image. **b** The projection of the HOPE-SIM image after 3D correlation with the FIB image. Microspheres corresponding to the green microspheres in (**a**) are tagged by white circles and crosses. **c** FIB image. Microspheres corresponding to the microspheres in (**a**) and (**b**) are tagged by white circles and crosses accordingly. **d** 3D correlation between HOPE-SIM and FIB images. Microspheres tagged with red circles and labeled from 1 to 9 correspond to the red microspheres in (**a**), respectively. **e** Corresponding position deviations of the nine red microspheres between (**a**) and (**d**), respectively. **f** HOPE-SIM image of a vitrified solution of fluorescent blue microspheres (green channel) mixed with polymethyl methacrylate beads that were dropped directly on the EM grid. The fluorescence image (green color) is merged with the bright-field image. Five microspheres shielded by polymethyl methacrylate beads are selected as targets and named as B1-5. **g** Cryo-FIB image at the target region. **h** Cryo-CLEM image at the region in (**g**). The selected microspheres in (**f**) are indicated accordingly. **i** Cryo-FIB fabrication of the target microspheres in (**h**). **j** After cryo-FIB fabrication, cryo-SEM image (SE signal) of the grid at the target region was captured and merged with the fluorescent image in (**f**). **k** High magnified cryo-SEM images (SE signal) of each lamella at the positions of B1-B5. The cross sections of both 1 μm fluorescent microspheres and bigger polymethyl methacrylate beads are clearly resolved. And the diameters of the cross sections of fluorescent microspheres can be measured directly from the SEM images and the resulted statistical distribution of the diameters is plotted in (**l**). See Supplementary Data 5 for raw cryo-FIB images of (**g**), (**h**), and (**j**).

selected mitochondrion can be precisely targeted by 3D-View for cryo-FIB milling, which was subsequently captured by cryo-ET (Fig. 1).

In addition to targeting mitochondria, we tried a second test with more biological implications to visualize virus assembly in situ from infected cells. Herpesviridae is a large family that consists of alpha, beta, and gamma-herpesviruses. Murine gammaherpesvirus 68 (MHV-68), like human gamma-herpesviruses, can establish a productive infection of fibroblast or epithelial cell lines derived from mammalian species and therefore is a model system for the study of pathogenic mechanisms associated with human

gammaherpesvirus infection. Herpesvirus maturation includes four stages: capsid formation in the nuclei, primary envelopment and passing through the nuclear membrane, tegumentation and secondary envelopment in the cytoplasm, and egressing from the cell[44]. Using conventional transmission electron microscopy of plastic-embedded sections, previous studies observed "tegument deposits" in the cytoplasm of MHV-68-infected BHK-21 cells, where capsids accumulate to acquire a series of tegument proteins[45].

Here, we labeled the capsid protein ORF65 of MHV-68 with the fluorescent tag mCherry (red channel) and utilized our HOPE-SIM-based cryo-CLEM workflow to investigate the native

in situ structure of the "tegument deposit" in ORF65-mCherry MHV-68-infected BHK-21 cells, which were cultured and vitrified on the EM grid (Fig. 4a–c). We used 1 μm blue microspheres as fiducial markers (green channel) to correlate 3D cryo-SIM and 2D cryo-FIB images in 3D-View and identified the tegument deposit location in the cytoplasm (Fig. 4d). Then, we performed site-specific cryo-FIB milling and prepared cryo-lamella with a thickness of 200 nm (Fig. 4e). We took a low-magnification cryo-SEM micrograph of the cryo-lamella, merged it with the corresponding cryo-SIM image, and confirmed the fluorescent signal of the target viral particles located in the cryo-lamella (Fig. 4f). Then the cryo-lamella was transferred for cryo-EM imaging (Fig. 4g) and cryo-ET reconstruction with a high magnification, where we found target viral particles that were under tegumentation (Fig. 4h and Supplementary Movies 3 and 4).

For the third test, we tried a more challenging target, namely the human centriole in HeLa cells. Human centrioles are closely related to tumorigenesis and multiple hereditary diseases[46,47]. There are only one or two pairs of centrioles in each mammalian cell, which greatly limits in situ study of their structures without an efficient cryo-CLEM workflow. Here, we used HeLa cells expressing the mCherry-tagged PACT (pericentrin-AKAP450 centrosomal targeting) domain[48], which is one of the centrosome components, to target and visualize the human centriole by cryo-ET (Fig. 5). Again, we used 1-μm blue microspheres as fiducial markers (green channel) to correlate the 3D cryo-SIM and 2D cryo-FIB images in 3D-View and identify the location of the centrosomes (Fig. 5a–d), which frequently appeared as two fluorescence dots in the cryo-SIM images (Fig. 5c). Then we performed site-specific cryo-FIB milling to prepare the target cryo-lamellae with a thickness of ~200 nm (Fig. 5e, f), which was subsequently verified by cryo-EM (Fig. 5g, h and Supplementary Fig. 8). Further reconstruction by cryo-ET revealed the 3D in situ structure of a pair of perpendicular centrioles (Fig. 5i–m and Supplementary Movies 5–8). Notably, according to our current statistics, we managed to fabricate 19 thick cryo-lamellae from 19 vitrified cells and put back for cryo-SIM verification and we found 16 cryo-lamellae retained the fluorescence signal of target centrosomes (Supplementary Fig. 9 and Supplementary Data 9), suggesting a high success targeting rate.

## Discussion

In situ structural study of macromolecules in their native cellular context by cryo-ET has become a new frontier of structural biology. Subsequently, the demand for improving the overall workflow, from specimen preparation and data collection to image analysis, has increased. Additionally, the current protocol of image correlation among cryo-FM, cryo-FIB and cryo-ET needs to be further improved with better efficiency and accuracy. The commonly used wide-field cryo-FM[13,21,22] or recently developed cryo-Airyscan confocal microscopy (CACM)[12] have limited resolution and lack 3D resolution, which hampers the correlation accuracy with the subsequent 2D image from cryo-FIB. Superresolution fluorescence microscopy under cryogenic conditions based on the photon-activated localization principle has been developed with a nanometer-scale resolution;[40,49] however, the high illumination power that is required increases the high risk of devitrification. Compared to conventional wide-field fluorescence microscopy, SIM enables a twofold improvement in the diffraction-limited resolution in three dimensions[28,34–38] with a relatively low illumination power (10–100 mW cm$^{-2}$). The wide-field nature of SIM makes it light-efficient and decouples the acquisition speed from the size of the lateral field of view, meaning that high frame rates over large volumes are possible[37]. Furthermore, SIM can be combined with a superresolution imaging algorithm[38] or a progressive deep-learning superresolution strategy[50,51] to achieve a higher

resolution. Therefore, cryo-SIM provides a promising solution to improve the accuracy of the current cryo-CLEM workflow.

In this study, we developed a HOPE-SIM-based cryo-CLEM workflow by incorporating the SIM module into our previously developed HOPE system to enable high-resolution 3D cryo-FM, upgrading the original vacuum system to further eliminate ice contamination/deposition, and writing HOPE-SIM View/3D-View software to control the hardware and perform image correlation semi-automatically. We demonstrated that our HOPE-SIM system can achieve cryo-SIM imaging with a resolution of ~200 nm in the lateral direction and ~500 nm in the z-direction. These results are better than the conventional wide-field cryo-FM and deconvolution modes. We verified the precision of the correlation between cryo-SIM and cryo-FIB as 110 nm, which is good enough to perform accurate site-specific cryo-FIB fabrication of cryo-lamella with a thickness of ~200 nm. We applied our HOPE-SIM-based cryo-CLEM workflow to successfully target MitoTracker-stained mitochondria in HeLa cells, visualize mCherry-tagged MHV-68 virions under tegumentation in infected BHK-21 cells, and obtain a tomogram of the human centrioles in HeLa cells. The high success rate of targeting the human centrioles suggests the robustness and accuracy of our cryo-CLEM workflow.

Compared to the published and commercially available non-integrated cryo-CLEM system (Supplementary Table 1), our HOPE-SIM-based cryo-CLEM system offers numerous advantages, including the use of a high-vacuum chamber with an embedded high NA objective lens kept at room temperature and an upgraded anti-contamination system to allow long-time high-resolution cryo-FM imaging without the concern of the lens becoming frozen/damaged and of ice growth; with the specially designed holder tip to accommodate the AutoGrid and commercial cryo-multi-holder, the cryo-specimen can be efficiently transferred from cryo-FM to cryo-FIB, and then to cryo-ET without touching the grid directly, which minimizes the risk of specimen damage during specimen transfer. With cryo-SIM imaging and the 3D correlation algorithm, we could achieve high-resolution fluorescence microscopy, especially with a better z-direction resolution, and then realize a precise 3D correlation with cryo-FIB 2D images, resulting in accurate site-specific cryo-FIB milling.

To be noted, besides the non-integrated cryo-CLEM systems, another cryo-CLEM concept has been recently developed, which integrates the fluorescence imaging system into the cryo-FIB/cryo-SEM chamber. These integrated systems include PIE-Scope[22], iFLM[52], METEOR[53], and the coincident three-beam microscope[54], as well as the very recently published ELI-TriScope[55] and CLIEM[56] by us and our colleagues. Besides avoiding specimen transfer during microscopies to minimize the risks of specimen damage and ice contamination, these systems can also achieve correlative imaging without the need of fluorescent fiducial markers after a proper pre-calibration[54–56], therefore significantly increasing the cryo-CLEM workflow efficiency. Our ELI-TriScope[55] provides another unique advantage to monitor the fluorescence of the target during cryo-FIB milling, therefore minimizing the off-target effect induced by cryo-FIB milling and accompanied with the cryo-lamella deformation. The design of our HOPE-SIM system is fully compatible with our ELI-TriScope system by utilizing the same cryo-multiholder to transfer cryo-specimen, therefore providing a complete solution for the future cryo-CLEM experiment.

Overall, our HOPE-SIM system-based cryo-CLEM workflow provides an efficient non-integrated solution to achieve accurate target cryo-FIB fabrication of cryo-lamella ready for site-specific cryo-ET study. Our HOPE-SIM system is versatile and can be adapted for various fluorescence and electron microscopes by utilizing proper stages, cryo-holders, and cartridges. In the future, besides further integration with our ELI-TriScope system[55] in one

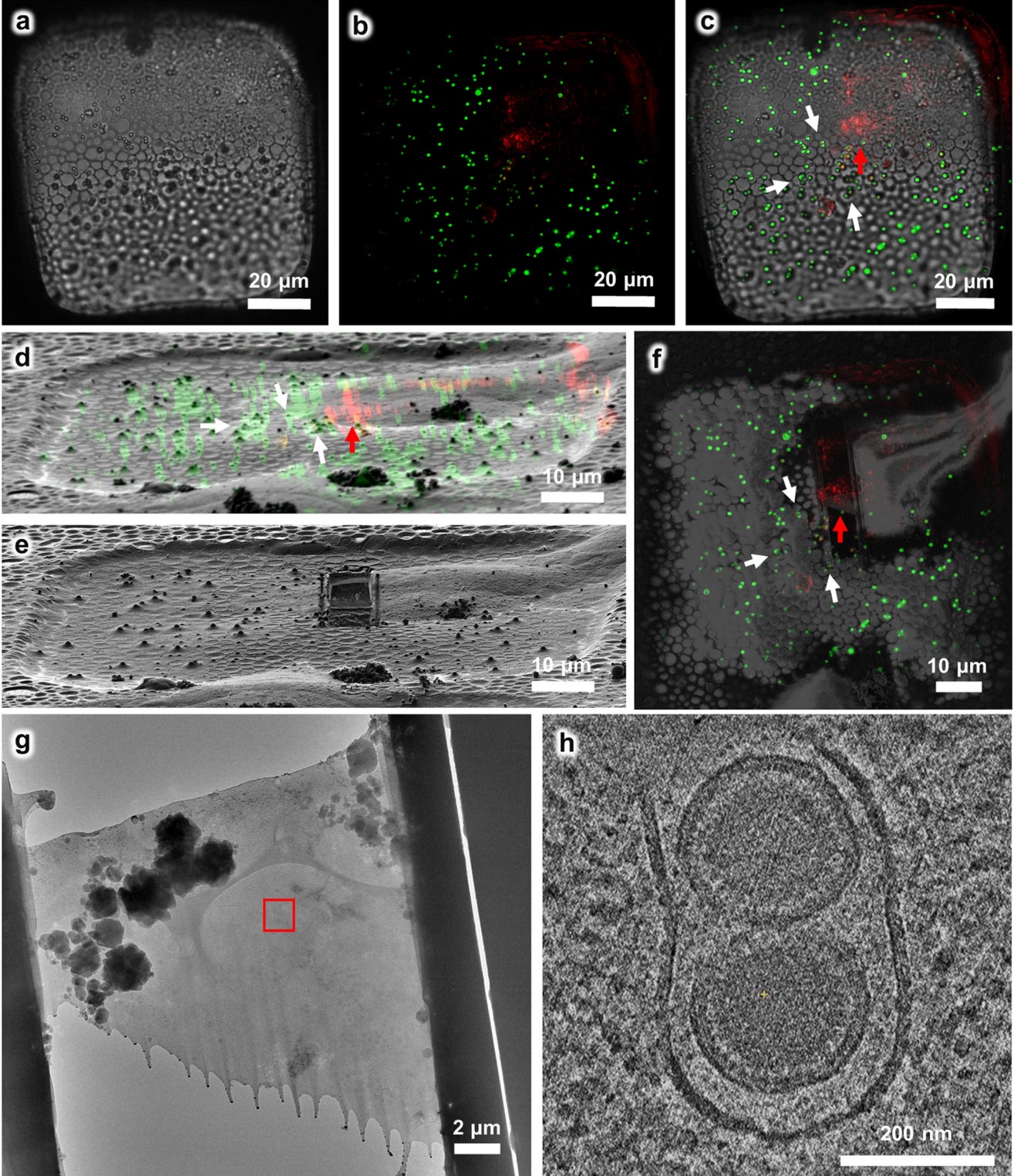

**Fig. 4 Using the HOPE-SIM-based cryo-CLEM workflow to capture the MHV-68 viral particles in host cells. a** Bright-field image of the target square. **b** The *z* projection of HOPE-SIM fluorescent image. Green, fluorescent microspheres. Red, MHV-68 virus. **c** The fluorescent image in (**b**) is merged with the bright-field image in (**a**) to show the location of target signal. **d** 3D correlation between the cryo-SIM and cryo-FIB images. **e** Cryo-FIB image of the target square after fabrication. **f** Cryo-SEM image of the target square after fabrication, which is merged with the *z* projection of cryo-SIM image in (**b**). **g** Cryo-EM micrograph (×3,600) of the cryo-lamella in (**f**). The target region marked by the red rectangle is subjected for cryo-ET reconstruction in (**h**) with a magnification of ×64,000, where one slice to the tomogram shows different types of viral particles that are under tegumentation. See Supplementary Data 6 for raw cryo-FIB images of (**d**) and (**e**), raw cryo-SEM image of (**f**), and raw cryo-EM image of (**g**).

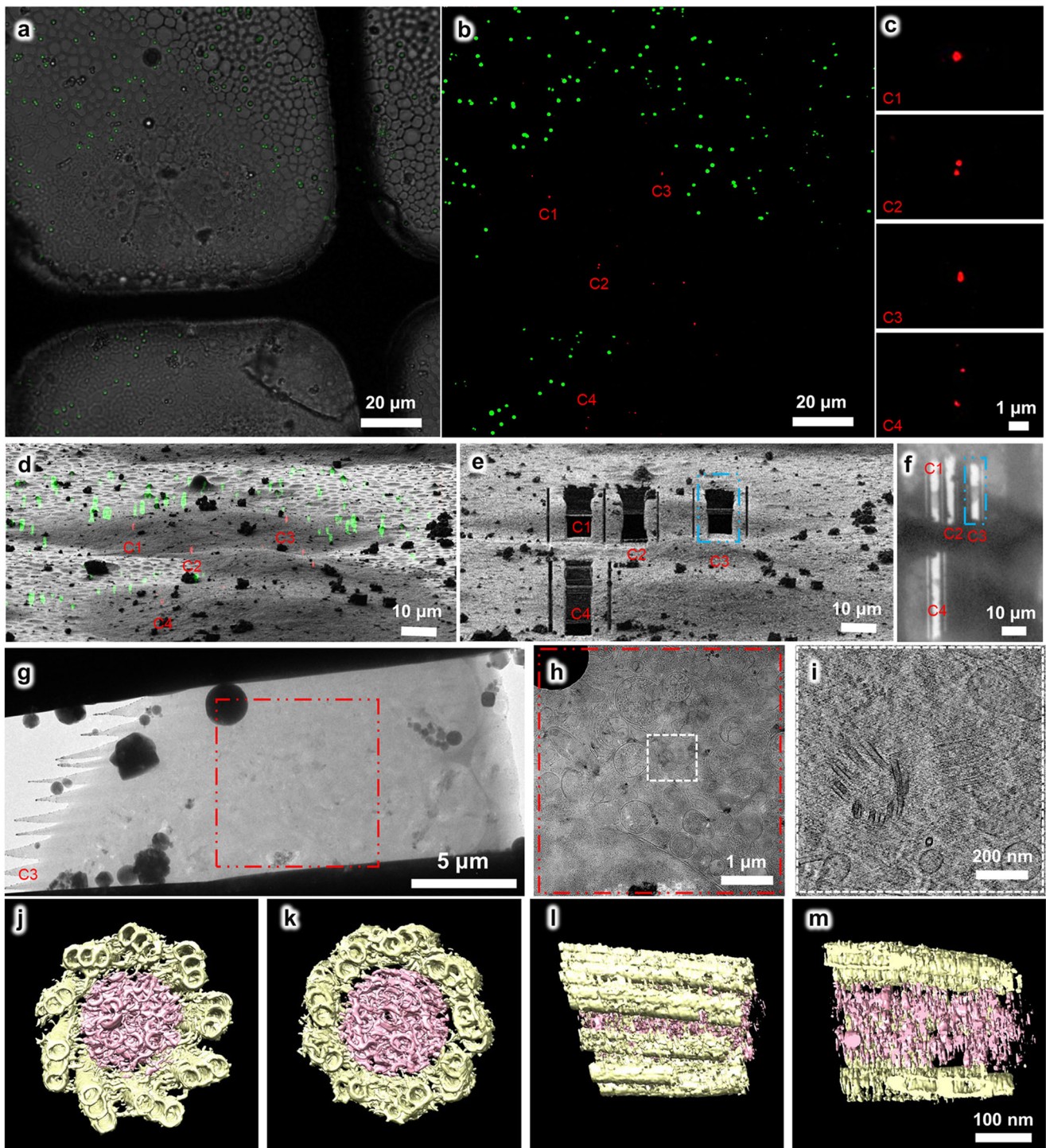

**Fig. 5 Using the HOPE-SIM-based cryo-CLEM workflow to capture the centrosomes in HeLa cells. a** Bright-field image of the target square is merged with the *z* projection of HOPE-SIM fluorescent image in (**b**), where the microspheres (green) are well spread, and the fluorescence-labeled centrosomes (red) are well resolved. Four sets (C1-4) of centrosomes are selected C1-4 for subsequent cryo-FIB fabrication. **c** Zoomed-in view of the four sets of centrosomes in (**b**). **d** 3D correlation between HOPE-SIM and cryo-FIB images to localize the positions of the selected centrosomes. **e** Cryo-FIB image after fabrication at the positions of C1-4 in (**d**). The cryo-lamella C3 indicated by the blue dashed rectangle is subjected to cryo-ET data collection. **f** Low-magnification (×155) cryo-EM image of the cryo-lamellae in (**e**). **g** Cryo-EM micrograph of the C3 cryo-lamella with the magnification of ×4300. **h** Cryo-EM micrograph (×8700) of the C3 cryo-lamella at the region of the red square in (**g**). The target region marked by the white square is subjected for cryo-ET data collection and reconstruction. **i** One slice of the tomogram (×53,000) of the target region, showing the target centrioles with one in the top view and another in the side view. **j–m** 3D in situ structure of the centriole in HeLa cells with different views, (**j**) top, (**k**) bottom, (**l**) side, and (**m**) cross. The structure is segmented using EMAN2[61] and rendered in Chimera[62]. The triplet tubules are shown in yellow, and the internal scaffold structures are shown in pink. To be noted, this structure was derived from another tomogram (×26,000) of cryo-lamella with a thickness of ~300 nm. See Supplementary Data 7 for raw cryo-FIB images of (**d**) and (**e**), and raw cryo-EM image of (**f**).

direction, we will also upgrade our HOPE-SIM system using the progressive deep-learning superresolution strategy to achieve a higher resolution and update the 3D-View software to realize fully automatic image correlation, making a more efficient and accurate non-integrated solution of site-specific cryo-FIB milling.

## Methods

**Cell culture and vitrification.** Indexed ultraviolet-sterilized grids with one quarter trimmed (D-grids, T11012SSF, TIANLD, China) were used for HeLa and BHK-21 cell culturing.

HeLa cells labeled by MitoTracker or expressing mCherry fused to the PACT (pericentrin-AKAP450 centrosomal targeting) domain[48] were seeded onto ultraviolet-sterilized D-grids and cultured in complete Dulbecco's modified Eagle's medium (DMEM) supplemented with 10% fetal bovine serum, 1% penicillin and streptomycin. After 48 h of culture, the grids were subjected to subsequent plunge freezing.

BHK-21 cells were cultured in complete DMEM supplemented with 10% fetal bovine serum, 1% penicillin and streptomycin. Culture media was routinely changed every 2 days by replacement of half the old medium with new medium. The ultraviolet-sterilized D-grids were placed into a 12-well plate culture dish before cell seeding (carbon film side upward). Then, BHK-21 cells were seeded. After cells attached to the D-grids, they were infected with ORF65-mCherry MHV-68 virus at a multiplicity of infection (MOI) of 100 for 1 h. Then, the inoculum was removed and replaced with fresh DMEM plus 10% fetal bovine serum. After 24 h of infection, the D-grids were subjected to plunge freezing.

Specifically, 1.5 μL of blue fluorescent microspheres with a diameter of 1 μm (430/465 nm, F13080, Thermo Fisher Scientific Corp., OR), used as the markers for correlation between the cryo-SIM 3D and cryo-FIB 2D images were diluted with PBS in a 1:400 ratio and added onto the D-grid with the side where the cells were growing before cryo-vitrification.

**Optical path of the HOPE-SIM system.** A schematic diagram of the optical path of the HOPE-SIM system is presented in Fig. 2a, and a picture of the real optical setup is shown in Supplementary Fig. 2a. We set up a laser combiner of three lasers with wavelengths of 405 nm (50 mW, OBIS 405LX, Coherent), 488 nm (200 mW, SAPPHIRE 488-200 CW, Coherent), and 561 nm (200 mW, SAP-PHIRE 561-200 CW, Coherent). The laser beam emitted from the laser combiner passes through an acousto-optic tunable filter (Amplitude modulator, AOTF, AOTFnC-400.650-TN & MPDS4C-B66-22-74.158-RS controller, Quanta Tech) and then expands to a diameter of 25 mm. Then, the beam passes through a phase-and-angle modulator consisting of a polarizing beam splitter cube (PBS, CCM1-PBS251, Thorlabs), an achromatic half-wave plate (HWP, AHWP10 M-600, Thorlabs), and a ferroelectric spatial light modulator (SLM, QXGA-3DM, Forth Dimension Displays) to generate patterned illumination with three alternative orientations (Supplementary Fig. 2b). These three patterned beams pass through another lens and are modulated at the diffraction plane using a customized mask consisting of a lightproof disc with a seven-hole aperture. Only diffractions at 0 and ±1 orders are selected to pass for each patterned beam. All three patterned beams are eventually focused at the back focal plane of the objective lens (Supplementary Fig. 2c). After passage through the objective lens, interference patterns (Moiré fringe) are generated with five phases (0, π/5, 2π/5, 3π/5, 4π/5) and three rotation angles (0°, 60°, and 120°). The period, orientation, and relative phase of this excitation pattern can be finely tuned by adjusting the SLM setup. For each orientation and phase of the excitation pattern, the resulting fluorescence signals are collected by the objective lens and recorded on a high-sensitivity sCMOS camera (Prime 95B, Photometrics). All lenses used in the system are chosen as achromatic doublets. Optical apertures and masks are used to improve the illumination quality.

**3D-SIM data collection and image reconstruction.** Raw data serials of 3D-SIM using a dry objective lens (NA 0.9) were collected with an increment of 250 nm along the z axis. For each z-slice, 15 images were acquired for five phases and three orientations to satisfy Nyquist–Shannon sampling. Each raw image has a pixel size of 120 nm and a field of view (FOV) of 1200 × 1200 pixels. Then, the final raw image stack has a voxel size of 120 × 120 × 250 nm. The exposure time for each raw image depends on the overall fluorophore intensity, and normally, the entire dataset was collected within an average time of 5–10 min to capture a field of view with the required z-depth of ~20 μm.

SIM image reconstructions were performed using SoftWoRx 6.1.1 (GE Healthcare) with the following settings: channel-specific optical transfer functions (OTFs), a Wiener filter constant of 0.010, a background with negative intensities discarded, and drift correction with respect to the first angle. OTFs were generated from point spread functions (PSFs) by SoftWoRx 6.1.1, and PSFs of the HOPE-SIM system at cryogenic temperatures were measured by using 200-nm fluorescent microspheres (405 nm/488 nm/560 nm, Thermo Fisher Scientific Corp., OR). For each z-slice, the final reconstructed image contains 2400 × 2400 pixels with a pixel size of 60 nm. The open-source software SIMcheck[43] was used to verify the performance of the SIM optical system, tune parameters and recognize artifacts.

**Development of 3D-View for image correlation.** We defined the coordinate vector of each fluorescent microsphere as $X = \begin{bmatrix} x \\ y \\ z \end{bmatrix}$; this provided two datasets of vectors, $X_{SIMi}(i = 1, \ldots, N)$ and $X_{FIBi}(i = 1, \ldots, N)$, for the selected corresponding fluorescent microspheres from the cryo-SIM and cryo-FIB images, respectively. The centroid of each dataset can be calculated as follows:

$$\mu_{SIM} = \frac{1}{N}\sum_{i=1}^{N} X_{SIMi} \quad (1)$$

$$\mu_{FIB} = \frac{1}{N}\sum_{i=1}^{N} X_{FIBi} \quad (2)$$

In this study, we defined the rotation operation $R(\theta)$ of three Euler angles $(\theta_x, \theta_y, \theta_z)$ as follows:

$$R_x(\theta_x) = \begin{bmatrix} 1 & 0 & 0 \\ 0 & \cos\theta_x & -\sin\theta_x \\ 0 & \sin\theta_x & \cos\theta_x \end{bmatrix} \quad (3)$$

$$R_y(\theta_y) = \begin{bmatrix} \cos\theta_y & 0 & \sin\theta_y \\ 0 & 1 & 0 \\ -\sin\theta_y & 0 & \cos\theta_y \end{bmatrix} \quad (4)$$

$$R_z(\theta_z) = \begin{bmatrix} \cos\theta_z & -\sin\theta_z & 0 \\ \sin\theta_z & \cos\theta_z & 0 \\ 0 & 0 & 1 \end{bmatrix} \quad (5)$$

The 3D coordinates of the fluorescent microspheres in the cryo-SIM volume can be translated into 2D coordinates on the coordinate plane of the cryo-FIB image as follows:

$$X'_{SIMi}(x'_{SIMi}, y'_{SIMi}) = P(z)R_x(\theta_x)R_y(\theta_y)R_z(\theta_z)(X_{SIMi} - \mu_{SIM}) \quad (6)$$

where $P(z)$ is the projection along the z-direction and the $Z$–$Y$–$X$ rotation transformation system is used.

In addition, the coordinates of the fluorescent microspheres in the cryo-FIB image are translated as follows:

$$X'_{FIBi}(x'_{FIBi}, y'_{FIBi}) = (X_{FIBi} - \mu_{FIB}) \quad (7)$$

According to Eqs. (6) and (7), the deviation of the correlation between the cryo-SIM and cryo-FIB datasets can be defined as follows:

$$X_{SD} = \sqrt{\frac{(X'_{SIM1} - X'_{FIB1})^2 + (X'_{SIM2} - X'_{FIB2})^2 + \cdots + (X'_{SIMn} - X'_{FIBn})^2}{n}} \quad (8)$$

Using the least square methods, the optimized Euler angle $(\theta_x, \theta_y, \theta_z)$ can be solved to minimize the correlation deviation $X_{SD}$. Finally, the cryo-SIM 3D volume data can be projected and merged with the 2D cryo-FIB image using Eq. (6). The target positions defined in the cryo-SIM image can be located precisely on the cryo-FIB image.

**3D correlative registration and cryo-FIB milling.** The clearly visible fluorescent microspheres around the target cell were selected manually from the cryo-SIM 3D and cryo-FIB images. The distribution of the selected fluorescent microspheres should be on both sides of the cell, especially in the y direction, for better correlation accuracy. The centers of the microspheres in the XY plane were determined first. To further optimize the correlation between two correlative images, the Z height of fluorescent microspheres could be manually adjusted according to the shapes of the fluorescence microspheres (shown in Supplementary Movie 2). For the cryo-FIB image, the center of the microsphere was difficult to recognize because the microsphere was normally embedded in the ice and appeared as a bump. The position of the top of the bump represents the position of the microsphere edge. Therefore, with this knowledge, the center of the microsphere in the cryo-FIB image could be determined by finding the position that was at half the diameter of the microsphere from the top of the bump along the y direction (FIB milling direction). After the first round of correlation, some microspheres with large deviations were removed due to the significant errors when determining their centers. The correlation parameters were then updated by optimizing the z axis position of each microsphere. All these image correlation operations were performed in 3D-View (Supplementary Fig. 5).

Cryo-lamellae were prepared by cryo-FIB milling using the FEI Helios NanoLab 600i or Aquilos-2 FIB-SEM microscope (Thermo Fisher Scientific Corp., OR). Target cells with clear fluorescent signals were selected according to the above correlation map. The grid was then coated with an organometallic platinum layer for 5 s at a 5 mm work distance. Cryo-lamellae with ~200 nm thickness was obtained by milling gradually using a 4-step Ga+ beam current of 0.43 nA, 0.23 nA, 80 pA and 40 pA at a stage angle of 13–22°.

**Cryo-electron tomography data collection and reconstruction**. The cryo-lamellae were loaded into an FEI Titan Krios G3i (Thermo Fisher Scientific Corp., OR.) that was equipped with a Gatan GIF K2 4k × 4k camera (Gatan, Inc., Warrendale, PA) and operated at 300 kV in low-dose mode. Tilt series were collected bidirectionally with SerialEM software[57] using a tilt range of −60° to 40° in 2° intervals, nominal magnification of ×33,000 or ×53,000, defocus of −4 ~−6 μm and total dose of ~150 e Å$^{-2}$. The tilt series were aligned and reconstructed with a weighted back-projection algorithm using IMOD[58] or reconstructed using ICON[59,60].

**Segmentation and visualization**. Image segmentation of the centriole tomogram was performed using EMAN2[61], and 3D rendering was performed using Chimera[62].

**Statistics and reproducibility**. All the measurements were repeated at least three times independently, which resulted similar results. The statistics profiles and figures were generated using LabVIEW, Origin or Excel. The sample size of each measurement was stated in the paper.

**Reporting summary**. Further information on research design is available in the Nature Portfolio Reporting Summary linked to this article.

# Data availability

The raw micrographs and raw statistics data in this study are provided as Supplementary Data. The raw tilt serials and reconstructed tomograms are available upon request.

# Code availability

The LabVIEW program HOPE-SIM View for controlling the device is hardware dependent, and the 3D-View program for image correlation is hardware-independent. These two programs are available at GitHub with the link, https://github.com/hilbertsun/HOPE-SIM.

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

## Acknowledgements

All cryo-CLEM, FIB, and ET work was performed in the Center for Biological Imaging (CBI, http://www.ibp.cas.cn/cbi/), the Institute of Biophysics (IBP), and the Chinese Academy of Sciences (CAS). We are grateful to Dr. Xiaojun Huang, Boling Zhu, Xujing Li, Lihong Chen for their help with cryo-EM data collection and Mr. Zeyang Li and Ms. Lulu Qin for their help with cryo-FIB fabrication. We would also like to thank Dr. Fulin Wang and Prof. Jianguo Chen from Peking University for their kindness in providing HeLa cells expressing mCherry-PACT, Prof. Wei Ji and Prof. Dong Li from IBP for their kind help with light microscopy, and Ms. Ziyan Wang and Dr. Yun Zhu from IBP for their kind help with cryo-ET data processing. This work was equally supported by grants from the Ministry of Science and Technology of China (2017YFA0504700 to G.J.), the Strategic Priority Research Program of Chinese Academy of Sciences (XDB37040102 to F.S.) and the National Natural Science Foundation of China (31830020 to F.S.). In addition, this work was supported by the Technological Innovation Program of the Chinese Academy of Sciences (29Y8CZ021001 to G.J.), the CAS Key Technical Support Personnel Project (29Y9CQ041 to G.J.), and the National Natural Science Foundation of China (31801199 to SL, 31801201 to X.J., and 81630059 to H.D.).

## Author contributions

G.J. and F.S. initiated and supervised the project. G.J. and S.L. designed and constructed the HOPE-SIM system with mentoring by W.X. G.J. wrote the software programs. S.L. performed the cryo-CLEM experiments and cryo-ET work. X.J., X.Z., and H.D. performed the cell culturing and vitrification experiments. X.J., G.J., and S.L. performed the cryo-FIB fabrication work. T.N., S.L., X.J., and C.Q. performed image processing. S.L., G.J., and F.S. analyzed the data and wrote the manuscript.

## Competing interests

The authors declare the following competing interests: Parts of this study (HOPE-SIM system) have been assigned a Chinese patent for invention (CN202110919044.4).
