## [Peer Review File · Communications Biology]

Reviewers' comments:

Reviewer #1 (Remarks to the Author):

Review

The paper by Li et al. describes a technical upgrade of their already published HOPE system [1]. The authors incorporated a 3D structured illumination fluorescence microscope to their high-vacuum optical platform. Moreover, the authors present a new software tool, to correlate the cryo-LM and EM data with high correlation precision. Herewith, the authors claim to improve the entire cryo-FIB/cryo-ET workflow, especially, if rare events are studied. To proof their points, a test system (fluorescent beads) and three model systems are studied.

Overall, the article is understandable and well written, but proofreading by a native speaker would be necessary to remove the last ambiguities.

Regarding the presented results, I have some major comments prior to publication

First, the authors claim to have a correlation precision of about 150nm. From my point of view, a detailed quantitative analysis is missing to accept such a statement. I would suggest to perform one of the two experiments to strengthen their claim:

1) Fluorescent beads, which are typically used as calibration standard, typically have a specified diameter with a given distribution ($d \pm \Delta d$). If one assumes, that the diameter is not modified during plunge freezing, the diameter of the milled slice of the bead can be used to determine if the fluorescent bead was milled at the center or not. Herewith, it would be easy to quantify the correlation precision in z.

2) If core-shell particles (e.g. [2]) are used for this experiment, the correlation precision can be determined by taking tomograms of the milled center of the particle. For example, if the fluorescent bead has a center with the size of 100 nm and an outer shell with the size of a few micron, the beads could be prepared as already described in the paper. After milling, the center of the bead could be imaged by tomography and the z-position of the center could be quantitatively analyzed to quantify the correlation precision.

From my point of view, such a quantitative analysis is necessary to specify a certain correlation precision.

Second, the authors claim that they developed a new software tool to correlate the cryo-LM and cryo-FIB data but from my point of view, the similarities to already published software is quite high (see for example [3] and [4]). At least, I was not able to see the major improvement of this software.

Third, the authors claim in the last part of their discussion that the vacuum system was upgraded to further eliminate ice contamination. From my point of view, such a statement can't be made since any analysis is missing. Moreover, the overview, which is presented in figure 5d shows quite some ice contamination (during transfer).

If water deposition was tackled by the upgraded vacuum system, the authors should quantify the impact of their developments like it was done in the paper by Tacke et al. [5].

Some minor points: the used references should be updated. Here are some suggestions to include [6-8].

References:

[1] Li, S. et al. High-vacuum optical platform for cryo-CLEM (HOPE): A new solution for non-integrated multiscale correlative light and electron microscopy. *Journal of Structural Biology* 201, 63-75 (2018).

[2] Jarzebski, M. et al. Core-shell fluorinated methacrylate nanoparticles with Rhodamine-B for confocal microscopy and fluorescence correlation spectroscopy applications. *Journal of Fluorine Chemistry* 183, 92-99 (2016)

[3] Wu, G. H. et al. Multi-scale 3D Cryo-Correlative Microscopy for Vitrified Cells. *Structure* 28, 1231-1237.e1233 (2020).

[4] Arnold, J. et al. Site-Specific Cryo-focused Ion Beam Sample Preparation Guided by 3D Correlative Microscopy. *Biophysical Journal* 110, 860-869 (2016).

[5] Tacke, S. et al. A streamlined workflow for automated cryo focused ion beam milling. *Journal of Structural Biology* 213, 107743 (2021)

[6] Robertson, M. J. et al. Drug discovery in the era of cryo-electron microscopy. *Trends in Biochemical Sciences* 47, 124-135 (2022)

[7] Wang, Z. et al. Structures from intact myofibrils reveal mechanism of thin filament regulation through nebulin. *Science* 375, 738 (2022)

[8] Gadadhar, S. et al. Tubulin glycylation controls axonemal dynein activity, flagellar beat, and male fertility. *Science* 371, 144 (2021)

Reviewer #2 (Remarks to the Author):

Author developed a HOPE-SIM system which contains a high vacuum stage and is compatible with FEI auto grid. HOPE-SIM achieves around 150 nm resolution in 3D and was successfully applied in 3 biological samples (MHV-68 infected cell and mitochondria and centrosomes in HeLa cells). HOPE-SIM increases the precision for cryo-FIB lamella fabrication in the Z direction.

Major Comment

1. Author should include bright field light microscopy and cryo-SEM images from all experiments because both bright field light microscopy and cryo-SEM images will provide more information about the sample including cell edges, carbon holes and ice contamination, which are useful for locating ROI areas and confirming the precision of HOPE-SIM system.
2. The fluorescent signals do not seem to fit the ion beam image well because the red signals don't align? (figure 4a). Author should provide individual images of SEM, ion beam and fluorescent before and after FIB which will help the reader understand the virus distribution in cell and confirm the precision of HOPE-SIM system.
3. Author should provide the cryo light microscopy image and cryo-SEM image in figure 5. In addition, the success rate here is not fair because the cryo-EM images are missing and cannot prove the lamella contains centrosomes.
4. At figure 2 b, the excitation pattern that authors are using. The fluorescence image that authors get (authors show below), should also look stripy, but it doesn't. The degree of stripyness is referred to in SIM as the modulation depth, and theirs is near zero so it really shouldn't work. Further, authors validate their SIM by showing that the fluorophores are hollow but fluorophores are not supposed to be. Fluorospheres are filled. The way they produce them is doping a swelled bead with dye and then shrinking it. Here is the reference <https://www.biorxiv.org/content/10.1101/2022.08.23.505029v1> , which was just published in *Journal of physical chemistry*.

Author's SIM might be a perfectly good widefield setup but is deeply flawed. It needs major revision.

Minor

1. Author should clearly label the 2D and 3D fluorescent images and indicate ROI areas in different stages in Figure 1.

2. Author should provide light microscopy and SEM images in Fig3, as well as SEM and ion beam images before milling.

Response to Reviewers

Reviewer #1:

The paper by Li et al. describes a technical upgrade of their already published HOPE system [1]. The authors incorporated a 3D structured illumination fluorescence microscope to their high-vacuum optical platform. Moreover, the authors present a new software tool, to correlate the cryo-LM and EM data with high correlation precision. Herewith, the authors claim to improve the entire cryo-FIB/cryo-ET workflow, especially, if rare events are studied. To proof their points, a test system (fluorescent beads) and three model systems are studied.

Overall, the article is understandable and well written, but proofreading by a native speaker would be necessary to remove the last ambiguities.

Response #1:

We would like to thank this reviewer for his/her professional comment and fruitful suggestions, which have helped us to improve the quality of our manuscript. The manuscript has been further polished with a careful proofreading.

Regarding the presented results, I have some major comments prior to publication

First, the authors claim to have a correlation precision of about 150nm. From my point of view, a detailed quantitative analysis is missing to accept such a statement. I would suggest to perform one of the two experiments to strengthen their claim:

1) Fluorescent beads, which are typically used as calibration standard, typically have a specified diameter with a given distribution ($d \pm \Delta d$). If one assumes, that the diameter is not modified during plunge freezing, the diameter of the milled slice of the bead can be used to determine if the fluorescent bead was milled at the center or not. Herewith, it would be easy to quantify the correlation precision in z.

2) If core-shell particles (e.g. [2]) are used for this experiment, the correlation precision can be determined by taking tomograms of the milled center of the particle. For example, if the fluorescent bead has a center with the size of 100 nm and an outer shell with the size of a few microns, the beads could be prepared as already described in the paper. After milling, the center of the bead could be imaged by tomography and the z-position of the center could be quantitatively analyzed to quantify the correlation precision.

From my point of view, such a quantitative analysis is necessary to specify a certain correlation precision.

Response #2:

Firstly, we would like to apologize for the ambiguity of our original statements. The correlation precision of our HOPE-SIM cryo-CLEM workflow is defined as the standard deviation of the coordinates of the targeted fluorescent microspheres between 3D cryo-SIM and 2D cryo-FIB images. In our revised manuscript, we computed the exact standard deviation as 110 nm, which defined the correlation precision of our system. The relevant manuscript has been revised as follows.

“We then measured the targeting precision of cryo-FIB guided by the 3D correlation information in 3D-View. In our first measurement (Fig. 3a-e), we spread two differently sized and colored fluorescent microspheres (Green/0.5 μm , 505/515 nm, F8813, Thermo Fisher Scientific Corp., OR., red/0.2 μm , 580/605 nm, F8810, Thermo Fisher Scientific Corp., OR.) onto a GraFuture RGO TEM Grid (GraFutureTM-RGO001A, Shuimu BioSciences Ltd., CN) and left it dry. This grid was used to measure the correlation precision between cryo-SIM and cryo-FIB images. We recorded both green and red channels of cryo-SIM images (Fig. 3a), selected the channel of green fluorescent microspheres to correlate the 3D cryo-SIM and 2D cryo-FIB images (Fig. 3b-d), and then calculated the standard deviation of the red fluorescent microspheres (see Equation 8 in Methods), resulting in a correlation precision of 110 nm (Fig. 3e & Supplementary Data 2).”

Second, we performed further experiments to measure the real precision of target cryo-FIB milling, especially in z direction, according to the first approach suggested by this reviewer. Figure 3 has been revised with the new data and two supplemental figures (Supplementary Figs. 6 and 7). And the relevant manuscript has been revised as follows.

“In our second measurement (Fig. 3f-l), to mimic the real experimental procedure of target cryo-FIB milling, we vitrified the mixed solution of 1 μm blue microspheres (430/465 nm, F13080, Thermo Fisher Scientific Corp., OR.) and 5 μm polymethyl methacrylate beads (PMMA5, Kai New Materials Ltd., CN) directly on the EM grid. With the shielding of big polymethyl methacrylate beads, many blue microspheres were embedded in the vitrified ice, mimicking the fluorescent target embedded in the vitrified cell. We then selected five embedded microspheres (B1-5 in Fig. 3f) to test the precision of the target cryo-FIB milling. First, we utilized the un-embedded blue microspheres that were discernible in cryo-FIB image to correlate the 3D cryo-SIM and 2D cryo-FIB

images (Fig. 3g-h). The correlation image (Fig. 3h) showed that the selected microspheres were unrecognizable and exactly close to the big polymethyl methacrylate beads according to the correlated fluorescent signals. Then cryo-FIB milling was performed at these target position, respectively (Fig. 3i, j). The resulted cryo-lamella with a thickness of ~200 nm was then imaged by SEM at a high magnification (Fig. 3k). The diameters of milled target microspheres were measured (Fig. 3l & Supplementary Fig. 6), resulting an averaged diameter measurement of 908 ± 106 nm (Supplementary Data 3). Compared to the statistical measurement of diameters of the blue microspheres in cryogenic condition (Supplementary Fig. 7 & Supplementary Data 4), which showed the averaged diameter of the blue microspheres was 997 ± 32 nm, the precision of target cryo-FIB milling in our workflow is at least smaller than 100 nm.”

Second, the authors claim that they developed a new software tool to correlate the cryo-LM and cryo-FIB data but from my point of view, the similarities to already published software is quite high (see for example [3] and [4]). At least, I was not able to see the major improvement of this software.

Response #3:

Thanks a lot for this comment. We agree with this reviewer that our 3D transformation algorithm is same with the previously published software. However, according to our experiences, we found the accuracy of the determination of the fluorescent geometric centers, especially the z coordinates, is important for the final correlation accuracy. Therefore, we would like to write a new software, 3D-View, to enable us to determine and optimize the z coordinates of the microspheres according to the cross-section of cryo-SIM 3D image.

To concentrate the novelty of our manuscript, we have moved the section “Development of 3D-View for image correlation” from RESULTS into METHODS in the revision.

Third, the authors claim in the last part of their discussion that the vacuum system was upgraded to further eliminate ice contamination. From my point of view, such a statement can't be made since any analysis is missing. Moreover, the overview, which is presented in figure 5d shows quite some ice contamination (during transfer).

If water deposition was tackled by the upgraded vacuum system, the authors should quantify the impact of their developments like it was done in the paper by Tacke et al. [5].

Response #4:

We would like to thank this reviewer for this important comment. It is important to distinguish the concepts between ice growth rate (water deposition as this reviewer mentioned) and ice contamination.

The upgrade of the vacuum system as well as the incorporation of the new anti-contamination system (ACS) are important to reduce the rate of ice growth (water deposition). In our revision, we have performed further experiments to measure the ice growth rate of specimen in the vacuum chamber of HOPE-SIM system and compared with our previous HOPE system (**Supplementary Fig. 4 & Supplementary Data 1, 8**). The result showed that the averaged ice growth rate in the vacuum chamber of HOPE-SIM system is 0.62% for 1 hr experiment and 1.3% for 8 hrs experiment. In our original HOPE system¹, the measured ice growth rate of the vacuum system was 3.9% per one hour, therefore the new vacuum system as well as the incorporation of the new anti-contamination system (ACS) of HOPE-SIM has resulted a significantly reduced ice growth rate.

In addition, we agree with this reviewer that the visible ice contaminations observed in some cryo-EM micrographs are most likely due to the cryo-transfer procedure. Therefore, to avoid any potential ambiguity, the relevant part in DISCUSSIONS has been revised as follows.

“Compared to the published and commercially available non-integrated cryo-CLEM system (**Supplementary Table 1**), our HOPE-SIM-based cryo-CLEM system offers numerous advantages, including the use of a high-vacuum chamber with an embedded high NA objective lens kept at room temperature and an upgraded anticontamination system to allow long-time high-resolution cryo-FM imaging without the concern of the lens becoming frozen/damaged and of ice growth; with the specially designed holder tip to accommodate the AutoGrid and commercial cryo-multi-holder, the cryo-specimen can be efficiently transferred from cryo-FM to cryo-FIB, and then to cryo-ET without touching the grid directly, which minimizes the risk of specimen damage during specimen transfer. ...”

Some minor points: the used references should be updated. Here are some suggestions to include [6-8].

References:

- [1] Li, S. et al. *High-vacuum optical platform for cryo-CLEM (HOPE): A new solution for non-integrated multiscale correlative light and electron microscopy. Journal of Structural Biology* 201, 63-75 (2018).
- [2] Jarzebski, M. et al. *Core-shell fluorinated methacrylate nanoparticles with Rhodamine-B for confocal microscopy and fluorescence correlation spectroscopy applications. Journal of Fluorine Chemistry* 183, 92-99 (2016)
- [3] Wu, G. H. et al. *Multi-scale 3D Cryo-Correlative Microscopy for Vitrified Cells. Structure* 28, 1231-1237.e1233 (2020).
- [4] Arnold, J. et al. *Site-Specific Cryo-focused Ion Beam Sample Preparation Guided by 3D Correlative Microscopy. Biophysical Journal* 110, 860-869 (2016).
- [5] Tacke, S. et al. *A streamlined workflow for automated cryo focused ion beam milling. Journal of Structural Biology* 213, 107743 (2021)
- [6] Robertson, M. J. et al. *Drug discovery in the era of cryo-electron microscopy. Trends in Biochemical Sciences* 47, 124-135 (2022)
- [7] Wang, Z. et al. *Structures from intact myofibrils reveal mechanism of thin filament regulation through nebulin. Science* 375, 738 (2022)
- [8] Gadadhar, S. et al. *Tubulin glycylation contraols axonemal dynein activity, flagellar beat, and male fertility. Science* 371, 144 (2021)

Response #5:

Thanks a lot for the help of this reviewer. Our manuscript has been further polished with a careful proofreading. The literatures suggested by this reviewer have been updated or added in the revision.

Reviewer #2 (Remarks to the Author):

Author developed a HOPE-SIM system which contains a high vacuum stage and is compatible with FEI auto grid. HOPE-SIM achieves around 150 nm resolution in 3D and was successfully applied in 3 biological samples (MHV-68 infected cell and mitochondria and centrosomes in HeLa cells). HOPE-SIM increases the precision for cryo-FIB lamella fabrication in the Z direction. Major Comment

1. Author should include bright field light microscopy and cryo-SEM images from all experiments because both bright field light microscopy and cryo-SEM images will provide more information about the sample including cell edges, carbon holes and ice contamination, which are useful for locating ROI areas and confirming the precision of HOPE-SIM system.

Response #6:

We would like to thank this reviewer for this important suggestion. In our revision, the corresponding bright field light microscopy and cryo-SEM images have been added in the revised Figs. 3, 4 and 5. The relevant raw images are also provided as Supplementary Data.

2. The fluorescent signals do not seem to fit the ion beam image well because the red signals don't align? (figure 4a). Author should provide individual images of SEM, ion beam and fluorescent before and after FIB which will help the reader understand the virus distribution in cell and confirm the precision of HOPE-SIM system.

Response #7:

We apologize for this ambiguity. We have repeated the experiments and revised Fig. 4 accordingly.

3. Author should provide the cryo light microscopy image and cryo-SEM image in figure 5. In addition, the success rate here is not fair because the cryo-EM images are missing and cannot prove the lamella contains centrosomes.

Response #8:

We are grateful for this important suggestion and critics. Figure 5 has been revised by adding the corresponding cryo-light microscopy and cryo-SEM images.

In this work, we did not use cryo-EM to measure the success rate of HOPE-SIM based cryo-CLEM workflow. We fabricated 19 thick (~700 nm) cryo-lamellae from 19 vitrified cells and put back for cryo-SIM verification. Then we found 16 cryo-lamellae retained the fluorescence signal of target centrosomes (**Supplementary Fig. 9 and Supplementary Data 9**), yielding a success rate of 84%.

We have found the off-target effect induced by cryo-FIB milling and accompanied with the deformation of cryo-lamella when we fabricated thin (~200nm) cryo-lamellae for cryo-EM imaging. This effect can be efficiently solved by our recent developed ELI-TriScope system (Nature Methods, 20(2):276-283, 2023). The combination of our HOPE-SIM and ELI-TriScope systems will provide a complete cryo-CLEM solution for the future *in situ* cryo-ET studies.

4. At figure 2 b, the excitation pattern that authors are using. The fluorescence image that authors get (authors show below), should also look stripy, but it doesn't. The degree of stripyness is referred to in SIM as the modulation depth, and theirs is near zero so it really shouldn't work.

Further, authors validate their SIM by showing that the fluoropheres are hollow but fluoropheres are not supposed to be. Fluorospheres are filled. The way they produce them is doping a swelled bead with dye and then shrinking it. Here it the referenece <https://www.biorxiv.org/content/10.1101/2022.08.23.505029v1> , which was just published in Journal of physical chemistry.

Author's SIM might be a perfectly good widefield setup but is deeply flawed. It needs major revision.

Response #9:

We would like to thank this reviewer for his/her professional critics and suggestions.

Yes, as this reviewer mentioned, structured illumination microscopy (SIM) introduces patterned illumination light on the imaging target, resulting in low-frequency interference fringes (moiré fringes) that carry structural information beyond the diffraction limit of the system. Thus, moiré fringes should appear clearly which represent information that has changed position in reciprocal space. In our study, the optical path of SIM system is carefully designed and aligned. All three patterned beams are eventually focused at the back focal plane of the objective lens (**Supplementary**

Fig. 2c). Furthermore, the open-source software SIMcheck was used to verify the performance of our cryo-SIM optical system (**Supplementary Fig. 2**).

First, we would like to apologize for the ambiguity of our original Fig. 2b where the compressed images were used. In the revised Figure 2, we have replaced the panel b using the raw images, showing clear stripy illumination patterns.

Second, as mentioned by this reviewer, the fluorospheres should be filled, as described in the recent publication (*The Journal of Physical Chemistry B*, doi: 10.1021/acs.jpcc.2c05939). Besides, we also performed experiments to use laser scanning confocal microscope equipped with Airyscan 2 mode (Zeiss LSM 980, Germany) to obtain a super resolution (~120nm) image of the blue fluorospheres used in our study. The results showed (see **Figure R1** below) that the fluorescence intensity inside the bead is almost half of that at the surface. Therefore, we agree that our original statement of “hollow feature” is not rigorous. However, it is still possible to discriminate the shell of a fluorosphere using a super resolution fluorescent image.

Figure R1. Snapshot of Airyscan images of 1 μm blue fluorospheres. Left, Airyscan image using 100X 1.45NA objective lens. Right, the intensity plot profile of yellow line in the left image.

To avoid any potential ambiguity, we have revised Figs. 2b and 2c, and the corresponding legends as follows.

“(b) Diagrammatic (top) and real (bottom) corresponding subimages of the cryo-SIM illumination patterns (black and white) on the fluorescence microspheres (blue). (c) Comparisons of fluorescent images of 1 μm microspheres among different imaging modes: wide-field (WF), deconvolution and cryo-SIM. It is clear that cryo-SIM yields the image with the best resolution in both the XY and YZ planes.”

Minor

1. Author should clearly label the 2D and 3D fluorescent images and indicate ROI areas in different stages in Figure 1.

Response #10:

We would like to thank this reviewer for this important suggestion to make our presentation clearer. We have revised Figure 1 to label fluorescent images with “Z projection 2D image” and “The projection of HOPE-SIM images” accordingly. At the same time, the ROI areas are also indicated correspondingly.

2. Author should provide light microscopy and SEM images in Fig3, as well as SEM and ion beam images before milling.

Response #11:

Thanks a lot. Figure 3 has been revised according to the reviewer’s suggestion.

References

- 1 Li, S. *et al.* High-vacuum optical platform for cryo-CLEM (HOPE): A new solution for non-integrated multiscale correlative light and electron microscopy. *J. Struct. Biol.* **201**, 63–75, doi:<https://doi.org/10.1016/j.jsb.2017.11.002> (2018).

REVIEWERS' COMMENTS:

Reviewer #1 (Remarks to the Author):

The authors have dealt with all my points carefully. From my point of view, there is nothing to be said against publishing the submitted article.

Reviewer #2 (Remarks to the Author):

The author dramatically improves figures and makes the manuscript well. I am satisfied and appreciate the authors' response to my suggestions and questions. I suggest accepting this manuscript because it will benefit the cryo-CLEM, cryo-FIB, and Cryo-ET field.